# State-space Models with Layer-wise Nonlinearity are Universal Approximators with Exponential Decaying Memory

**Shida Wang**
Department of Mathematics
National University of Singapore
e0622338@u.nus.edu

**Beichen Xue**
Department of Statistics and Data Science
National University of Singapore
e0773769@u.nus.edu

## Abstract

State-space models have gained popularity in sequence modelling due to their simple and efficient network structures. However, the absence of nonlinear activation along the temporal direction limits the model's capacity. In this paper, we prove that stacking state-space models with layer-wise nonlinear activation is sufficient to approximate any continuous sequence-to-sequence relationship. Our findings demonstrate that the addition of layer-wise nonlinear activation enhances the model's capacity to learn complex sequence patterns. Meanwhile, it can be seen both theoretically and empirically that the state-space models do not fundamentally resolve the issue of exponential decaying memory. Theoretical results are justified by numerical verifications.

## 1 Introduction

State-space model [1–5] is an emerging family of neural networks specialized in learning long sequence relationships. It achieves significantly better performance compared with attention-based transformers in the long range arena (LRA) dataset [6, 7]. Despite its effectiveness, the state-space model is built on a relatively simple foundation of linear-RNN-like layers. One of the key advantages of state-space models is their simple recurrence, which enables efficient acceleration. In fact, this recurrence allows for an asymptotic computational complexity of only $O(T \log T)$, which is significantly better than the $O(T^2)$ complexity of traditional full-attention approaches [8]. A natural question would be whether SSM achieves this speedup with certain sacrifices in model capacity or memory property. It is currently unclear whether the state-space model's linear architecture with layerwise nonlinearity possesses sufficient expressive capacity to approximate any target sequence-to-sequence relationship. This knowledge would be important to answer pertinent questions regarding the model's ability to handle the complexity of real-world datasets characterized by diverse and intricate sequence relationships. In particular, considering the speed advantage of SSM over attention-based transformers, the universal approximation property impacts whether a state-space model could be a suitable replacement for a transformer.

In this paper, we study the universality of state-space model. Furthermore, the memory property is investigated and we show that state-space models also have an asymptotically exponential decaying memory.

The main contributions can be summarized as follow

1. We give a constructive proof for the universal approximation property for multi-layer state-space models. The width dependency on sequence length is analyzed.

37th Conference on Neural Information Processing Systems (NeurIPS 2023).

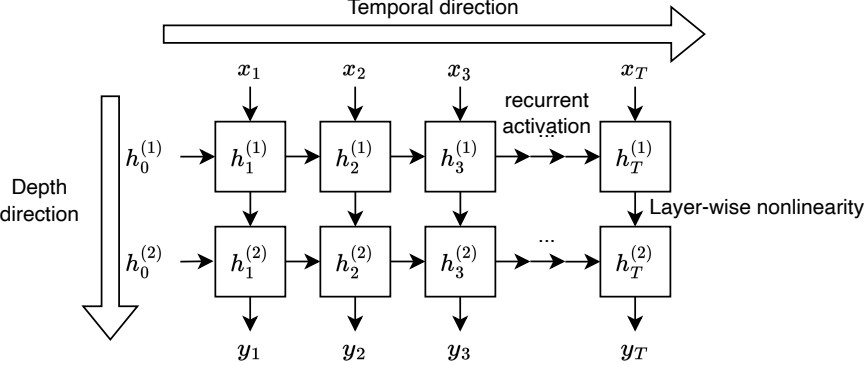

Figure 1: Network structure of two-layer state-space model.

2. The state-space models are shown to have an exponentially decaying memory, which coincides with usual recurrent neural networks.

3. Numerical verifications for the memory property are given on synthetic datasets.

## 2  Background

In this section, we introduce the general form of the state-space models. The classical results on universal approximation for recurrent neural networks are summarized. Based on the approximation result, the well-documented challenge of learning long-term memory via recurrent networks is summarized. In particular, we give the definition of memory function which we shall use in the derivation of memory decay.

### 2.1  State-space models

Single-layer state-space model can be viewed as a recurrent neural network without nonlinear recurrent activation. As is shown in Figure 1, the discrete time version of SSM [9] is

$$h_{k+1} = Wh_k + Ux_{k+1}, \qquad h_0 = 0 \tag{1}$$

$$y_k = Ch_k + Dx_k = CW^k h_0 + \sum_{i=1}^{k} CW^{k-i} Ux_i + Dx_k. \tag{2}$$

where $x \in \mathbb{R}^{d_{in}}, y \in \mathbb{R}^{d_{out}}, h \in \mathbb{R}^m, W \in \mathbb{R}^{m \times m}, U \in \mathbb{R}^{m \times d_{in}}, C \in \mathbb{R}^{d_{out} \times m}, D \in \mathbb{R}^{d_{out} \times d_{in}}$. Notice that we drop the bias term for simplicity. For multi-layer state-space model, the nonlinear activation is added layer-wise.

The continuous time version of a single layer is

$$\frac{dh_t}{dt} = Wh_t + Ux_t, \qquad h_0 = 0 \tag{3}$$

$$y_t = Ch_t + Dx_t = \int_0^T Ce^{W(t-s)} Ux_s ds + Dx_t. \tag{4}$$

It can be seen in Equation (4) that the first component of output $y$ is the convolution between kernel function $\rho(t) = Ce^{W(t-s)}U$ and $x$. In discrete case, the convolution operator $*$ is represented by

$$y_{[0,T]} = \rho(t) * x_{[0,T]} \Rightarrow y_k = \sum_{i=0}^{k} \rho_{k-i} x_i. \tag{5}$$

Compared with temporal convolution network (TCN) [10], which has a finite kernel size, state-space models can be regarded as implementing global convolution ($\rho(t) = Ce^{W(t-s)}U, t \geq 0$) over the

temporal axis. Numerically, since the kernel size is the same as the sequence length, the convolution implemented via fast Fourier transform can lead to significant accelerations [11]. Compared with attention-based transformer, it is shown that SSM can speed up the training 100x in learning sequences with length 64K [12]. The state-space model's temporal efficiency is achieved by eliminating the nonlinear activation function in its recurrent layers $h_{k+1} = Wh_k + Ux_k + b$, resulting in faster processing of sequential data via parallel scan [11].

## 2.2 Universal approximation in RNN

It is long-known that recurrent neural networks with nonlinear sigmoidal activations (such as tanh and sigmoid) are universal approximators.

**Theorem 2.1** (Simplified universality statement). *For any sequence to sequence map:* $\mathbf{H} : \mathbf{x} \to \mathbf{y}$ *and tolerance $\epsilon$, there exists a hidden dimension $m$ and weights $c, W, U, b$ such that the RNN with weights $(c, W, U, b)$ can approximate the target sequence to sequence map:*

$$\|y - \hat{y}\| \le \epsilon. \tag{6}$$

*where prediction sequence $\hat{y}$ is given by*

$$h_{k+1} = \sigma(Wh_k + Ux_k + b), \tag{7}$$

$$\hat{y}_k = C^\top h_k. \tag{8}$$

Universal approximation establishes the feasibility of learning sequence to sequence relationships via recurrent neural networks. However, typical approximation rate results depend on the sequence length [13]. Sequence length dependent approximation rate does not generalize to the case of sequence of infinite length. In learning sequences with infinite length, Li et al. [14] shows that linear RNNs have difficulty in learning non-exponential decaying memory. Various numerical experiments [15] confirm that adding nonlinear recurrent activation does not fundamentally change the decay. In state-space models, the nonlinearity is included in a layer-wise approach. It is unknown whether such layer-wise nonlinearity alone is sufficient to approximate any sequence to sequence relationships.

## 2.3 Memory function and curse of memory

In this paper we study the memory property of state-space model. Before we introduce the main results, we present a simple memory function definition in sequence modelling. Li et al. [14] proves that a bounded causal continuous regular time-homogeneous linear functional has the following Riesz representation:

$$y_t = H_t(\mathbf{x}) = \int_{-\infty}^{t} \rho(t - s)x_s ds. \tag{9}$$

Here $\rho : \mathbb{R}^+ \to \mathbb{R}$ is an $L_1$-integrable function. If $\rho_t$ rapidly decreases with $t$, then the target sequence relationship has a short-term memory.

Since $\rho_t$ fully captures the memory property of a linear functional [14], we call it the **memory function** (of the linear functional). In particular, when approximating the linear functional with linear RNNs, the model's memory function is $\hat{\rho}(t - s) = C^\top e^{W(t-s)}U$.

The **curse of memory** refers to the phenomenon that if a target linear functional can be approximated by a sequence of linear RNNs, the memory function $\rho_t$ decays exponentially.

Take the test input to be $\mathbf{x}_{\text{test}} = \begin{cases} 1 & t \ge 0, \\ 0 & t < 0. \end{cases}$ Notice that the derivative of the linear functional at test input extracts the memory function $\left| \frac{d}{dt} H_t(\mathbf{x}_{\text{test}}) \right| = |\rho(t)|_2$. Therefore a natural extension of the memory function [16] to the bounded causal continuous regular time-homogeneous nonlinear functionals will be:

$$\hat{\rho}(t) = \left| \frac{d\hat{y}_t}{dt} \right|_2, \quad \hat{y}_t = \widehat{\mathbf{H}}_t(\mathbf{x}_{\text{test}}). \tag{10}$$

It can be seen this definition is compatible with the memory function definition of linear functional. Also, this memory function can be evaluated by computing the models' derivative at the test input using finite difference method.

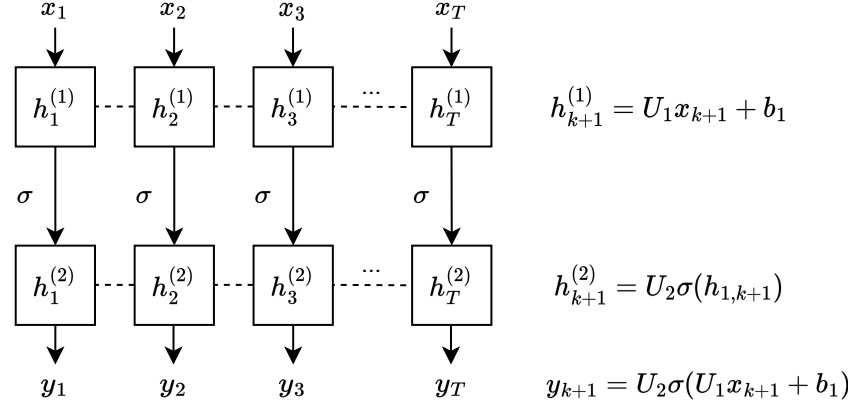

$$h_{k+1}^{(1)} = U_1 x_{k+1} + b_1$$

$$h_{k+1}^{(2)} = U_2 \sigma(h_{1,k+1})$$

$$y_{k+1} = U_2 \sigma(U_1 x_{k+1} + b_1)$$

Figure 2: Two-layer state-space model can approximate any continuous element-wise function

## 3   Main results

In this section, we first give a simple constructive proof to show that any element-wise function on input sequence can be approximated by a two-layer state-space model. Next, we show any temporal convolution can be approximated by a state-space model. The constructive proof for general nonlinear sequence to sequence functional is given based on the above propositions. Moreover, as the Kolmogorov-Arnold-representation-based construction has a weight number dependent on sequence length, it can be highly inefficient to construct a shallow wide network to learn long sequences. To reduce the widths dependency on the sequence length, a Volterra-series-based construction is demonstrated.

### 3.1   Two-layer SSM approximates element-wise function

By element-wise function, we mean learning sequence relationship with form $(x_1, \ldots, x_T) \rightarrow (f(x_1), \ldots, f(x_T))$.

**Proposition 3.1.** *For any given continuous function $f$ over a compact set $K$, let the activation $\sigma$ be sigmoidal function.*

$$\lim_{z \to \infty} \sigma(z) = 1, \quad \lim_{z \to -\infty} \sigma(z) = -1. \tag{11}$$

*There exists a sequence of two-layer state-space model with weights $\{W_1, W_2, U_1, U_2, b_1, b_2\}$ that can approximate sequence relationship $\mathcal{H} : (x_1, \ldots, x_T) \rightarrow (y_1, \ldots, y_T)$.*

$$\sup_t \sup_{\mathbf{x}} |f(\mathbf{x}) - \hat{\mathbf{y}}| \leq \epsilon. \tag{12}$$

*Here the two-layer state-space model is constructed by*

$$h_{k+1}^{(1)} = W_1 h_k^{(1)} + U_1 x_k + b_1, \tag{13}$$

$$h_{k+1}^{(2)} = W_2 h_k^{(2)} + U_2 \sigma(h_{k+1}^{(1)}) + b_2 \tag{14}$$

$$y_k = h_k^{(2)}. \tag{15}$$

The proof is included in Appendix B.2. The main idea is to approximate element-wise function $f(x)$ with $U_2 \sigma(U_1 x + b_1)$. A graphical demonstration is given in Figure 2. It can be seen the two-layer state-space model can approximate any element-wise function by setting $W_1 = W_2 = b_2 = 0$.

### 3.2   SSM approximates temporal convolution

In this part we show that state-space models can approximate temporal convolution: $y_k = \sum_{i=1}^{k} \rho_{k-i} x_i, \quad 1 \leq k \leq T$. According to Equation (3), the hidden state of SSM is the convolution between input sequence and exponentially decaying function $\hat{\rho}_k = CW^k U$. $y_k = \sum_{i=1}^{k} CW^{k-i} U x_i + CW^k h_0 + D x_k$.

Therefore the approximation problem of temporal convolution by state-space models is reduced to the approximation problem of general convolution kernel $\rho_k, 1 \leq k \leq T$ by exponentially decaying convolution kernel $\hat{\rho}_k, 1 \leq k \leq T$. The optimal weights are defined by $C, W, U = \arg\min_{C,W,U} \sup_k |\rho_k - CW^kU|$.

**Proposition 3.2.** *For any given convolution kernel $\rho_k, 1 \leq k \leq T$, the single-layer state-space model is universal approximator for temporal convolution.*

*In other words, for any $\epsilon > 0$, there exists a hidden dimension $m$ and corresponding weights $C, W, U$ such that $\rho_k = CW^kU$ satisfies*

$$\sup_k |\rho_k - \hat{\rho}_k| < \epsilon. \tag{16}$$

See Appendix B.3 for the proof. The main idea is to represent the single-layer state-space model output as a convolution between input and kernel function. The approximation of temporal convolution is then reduced to the approximation of general kernel with the SSM-induced kernels.

*Remark* 3.3. Although the Proposition 3.2 indicates that we can use single-layer state-space model to approximate any convolution, it does not reveal the necessary hidden dimension $m$ for such approximation.

## 3.3 Universality of SSM

Now we show that five-layer state-space model is universal. Without loss of generality, assume the output $y$ is one-dimensional. The main proof is based on the famous Kolmogorov-Arnold representation theorem [17]. By Kolmogorov-Arnold representation theorem, we know any multivariate continuous function $f : \mathbb{R}^d \to \mathbb{R}$ can be **represented** by

$$f(x_1, \ldots, x_d) = \sum_{q=0}^{2d} \Phi_q \left( \sum_{p=1}^{d} \phi_{q,p}(x_p) \right). \tag{17}$$

*Remark* 3.4. George Lorentz [18] shows that we can use the same $\Phi$: $f(x_1, \ldots, x_d) = \sum_{q=0}^{2d} \Phi \left( \sum_{p=1}^{d} \phi_{q,p}(x_p) \right)$. Sprecher [19] proves that it can be further reduced to the same $\phi$: $f(x_1, \ldots, x_d) = \sum_{q=0}^{2d} \Phi \left( \sum_{p=1}^{d} \lambda_p \phi(x_p + \eta p) + c_q \right)$. Braun and Griebel [20] gives the first constructive proof for the superposition. Moreover, the inner function $\phi$ is shown to be **independent** of target function $f$. It means the learning of function $\phi$ can be approximated without retraining for different target functionals.

We summarize the Kolmogorov-Arnold theorem as follow:

**Theorem 3.5** ([17, 20]). *Fix dimension $d \geq 2$. There are real numbers $a, b_p, c_q$ and a continuous and monotone function $\phi : \mathbb{R} \to \mathbb{R}$, such that for any continuous function $f : [0, 1]^d \to \mathbb{R}$, there exists a continuous function $\Phi : \mathbb{R} \to \mathbb{R}$ with*

$$f(x_1, \ldots, x_d) = \sum_{q=0}^{2d} \Phi \left( \sum_{p=1}^{d} b_p \phi(x_p + qa) + c_q \right). \tag{18}$$

**Proposition 3.6.** *For any bounded causal continuous sequence to sequence relationship $H : \{x_k\}_{k=1}^{T} \to \{y_k\}_{k=1}^{T}$ and tolerance $\epsilon > 0$, there exists a hidden dimension $m$ and corresponding state-space model (as constructed in Figure 3) such that the error of approximation*

$$|y_k - \hat{y}_k| \leq \epsilon, \quad k \in \{1, \ldots, T\}. \tag{19}$$

See the proof in Appendix B.4. The main idea is demonstrated in Figure 3, the nonlinear functions are separately approximated by two-layer state-space model.

*Remark* 3.7. The Kolmogorov theorem provides a construction for achieving universality in a five-layer state-space model. However, the quantity of hidden neurons increases linearly with the sequence length. This can become exceedingly burdensome when the sequence length escalates.

Another approach is from the perspective of Volterra Series, which features the sequence-length independent neurons.

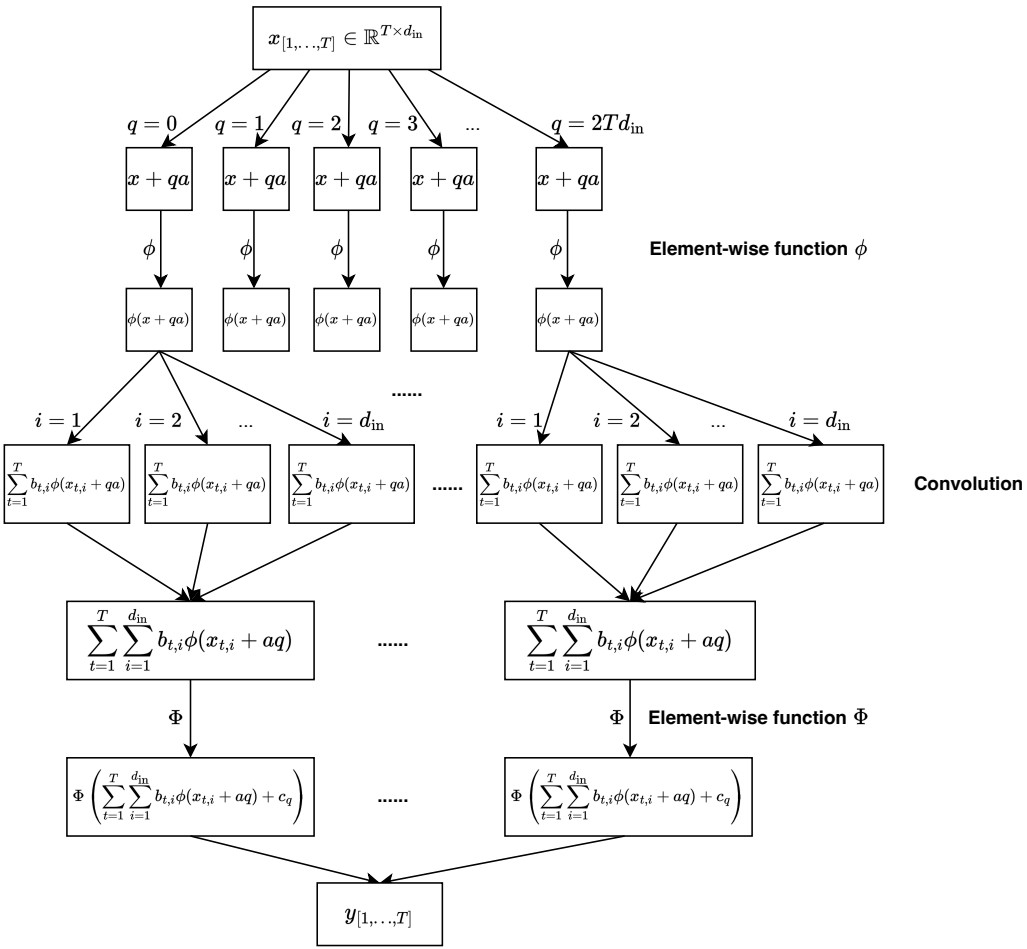

Figure 3: Multi-layer state-space models are universal approximators: Drawing from the Kolmogorov-Arnold representation theorem, we have $f(x_1, \ldots, x_d) = \sum_{q=0}^{2d} \Phi\left(\sum_{p=1}^{d} b_p \phi(x_p + qa) + c_q\right)$. Here, both element-wise nonlinear functions $\phi$ and $\Phi$ can be approximated by a two-layer state-space model, as shown in Proposition 3.1. Additionally, the temporal convolution is represented by a single-layer state-space model, as detailed in Proposition 3.2.

**Theorem 3.8** ([21]). *For any continuous time-invariant system with $x(t)$ as input and $y(t)$ as output can be expanded in the Volterra series as follow*

$$y(t) = h_0 + \sum_{n=1}^{N} \int_0^t \cdots \int_0^t h_n(\tau_1, \ldots, \tau_n) \prod_{j=1}^{n} x(t - \tau_j) d\tau_j. \tag{20}$$

*In particular, we call the expansion order $N$ to be the series' order.*

A simplified interpretation of the $N$-th Volterra Series expansion is the "$N$-th Taylor expansion in the sequence variable $\mathbf{x}$".

**Proposition 3.9.** *For any bounded causal continuous time-homgeneous sequence to sequence relationship $H : \{x_k\}_{k=1}^{T} \to \{y_k\}_{k=1}^{T}$ and tolerance $\epsilon > 0$, there exists a hidden dimension $m$ and corresponding state-space model (as constructed in Figure 4) such that the error of approximation*

$$|y_k - \hat{y}_k| \leq \epsilon, \quad k \in \{1, \ldots, T\}. \tag{21}$$

*Moreover, the neurons of the state-space model do not explicitly depend on the sequence length $T$.*

See the proof in Appendix B.5. The main idea is to approximate the convolution kernels $h_n(\tau_1, \ldots, \tau_n)$ by the low-rank tensor product of first-order convolution kernel.

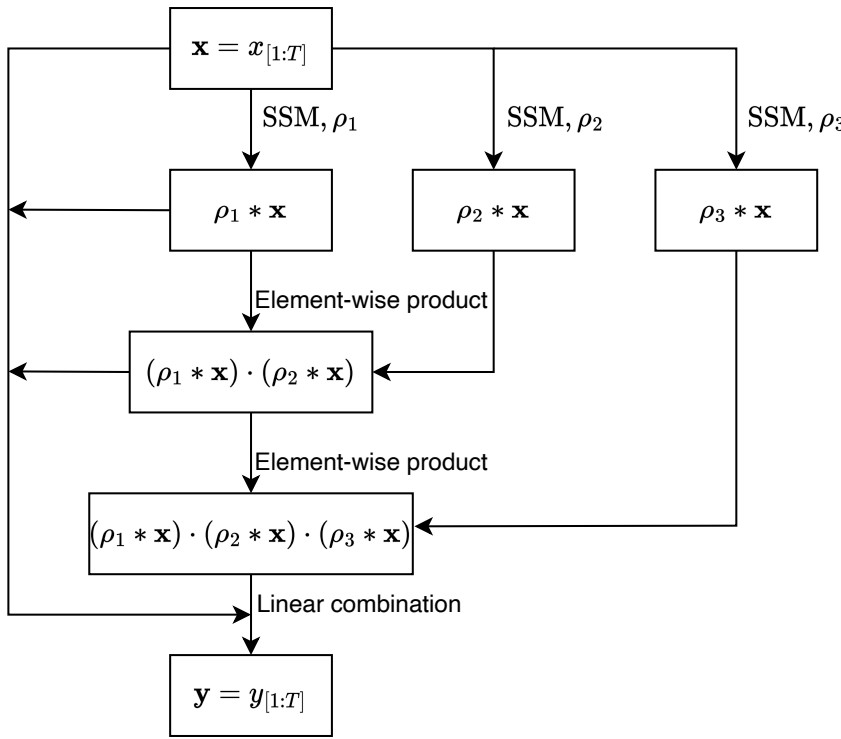

Figure 4: Volterra-series-type construction for state-space models

*Remark* 3.10. The advantage of Volterra-series-type construction is the approximation of the convolution kernel with SSM-induced kernel as well as the approximation of multivariable convolution kernel with tensor product of one-dimensional convolution kernel do not explicitly depend on the sequence length. Similar approaches have been adopted by idea of implicit convolution in CKConv [22].

## 3.4 Memory decay of SSM

In the ensuing discourse, we turn our focus towards an examination of the memory property inherent in state-space models. It has been thoroughly studied in literature that recurrent neural networks exhibit a phenomenon of exponential memory decay [14]. An intriguing question that naturally arises in this context is whether state-space models are plagued by similar challenges. Upon careful investigation, it is concluded that state-space models, much like their neural network counterparts, do possess an asymptotically exponential decaying memory.

**Proposition 3.11.** *Assume there exists a constant $c_0 > 0$ such that*

$$\lim_{t \to \infty} e^{c_0 t} \|x_t - x^*\| \to 0, \quad x^* := \lim_{t \to \infty} x_t. \tag{22}$$

*Assume the output $y_t$ is the output of single-layer state-space model with parameters $C, W, U$. Then, for the same constant $c_0$, if the output's derivative satisfies $\frac{dy_t}{dt} \to 0$*

$$\lim_{t \to \infty} e^{c_0 t} \|y_t - y^*\| \to 0. \tag{23}$$

At the same time, general smooth nonlinear activation does not change the exponential decay property of a sequence:

**Proposition 3.12.** *Assume there exists a constant $c_0 > 0$ such that*

$$\lim_{t \to \infty} e^{c_0 t} \|x_t - x^*\| \to 0, \quad x^* := \lim_{t \to \infty} x_t. \tag{24}$$

*For given Lipschitz continuous layer-wise activations $\sigma$, there exists a positive constant $c_0$ such that the output memory function $\frac{dy_t}{dt} \to 0$*

$$\lim_{t \to \infty} e^{c_0 t} \|y_t - y^*\| \to 0. \tag{25}$$

See the proofs for above two propositions in Appendix B.6 and Appendix B.7.

Based on the above two propositions, by induction we have the following theorem:

**Theorem 3.13.** *Assume* **H** *is a multi-layer state-space model with Lipschitz continuous function as the layer-wise activations. Assume the state-space model is stable in the sense that matrix $W$'s eigenvalue are bounded by 1.*

*There exists a positive constant $c_0$ such that the memory function (defined in Equation* (10)*) of state-space model is decaying exponentially*

$$\lim_{t \to \infty} e^{c_0 t} \hat{\rho}(t) \to 0. \tag{26}$$

## 4    Numerical verifications

Based on the generalization of memory function $\rho$ in linear functional, we verify the asymptotic memory decay of state-space models with simple randomly generated models. The definition is given in one-dimensional case, but it can be generalized to multi-variable case by taking different unit inputs in various coordinates. The motivation for the above definition is based on the idea of measuring the earlier input at the later output.

In our experiment, we construct various RNN models and SSM with random generated weights. It can be seen in Figure 5 that the memory of naive state-space models also has an exponentially decay memory. It is consistent with the previous theorem that state-space model has an asymptotic exponentially decaying memory. Notice that here the naive SSM is simply adding a tanh activation across layers without specially tuning the weights. Such random initialization can expose the memory issue more significantly as the S4 layer is constructed with several parameterization techniques. However, the manually constructed S4 still has an asymptotic exponential decaying memory as is shown in Figure 6.

## 5    Related Work

In this section, we introduce the previous works on state-space models. As the single-layer state-space model is a linear RNN, we summarize the related approximation work on RNN. In particular, the approximation result and memory result is emphasized as this paper works on the universal approximation property and memory decay property of SSM.

**State-space models**    State-space models originate from the HIPPO matrix which is optimal in the online function approximation sense [2–4]. The Hippo matrix initialization for recurrent matrix $W$ enables the state-space model to have a slow decaying memory. The universal approximation idea is heuristically demonstrated in Orvieto et al. [23]. However, the proof from the Koopman theory perspective only guarantees the existence of universal approximation. Our proof is a constructive proof which can be further generalized to study the approximation rate with respect to the hidden dimensions and network depths.

**Recurrent neural networks**    Recurrent neural networks (RNNs) [24] are one of the most popular neural networks for sequence modelling. Various results have been established in RNNs approximation theory, see Sontag [25], Hanson et al. [13]. Apart from the universal approximation, the exponential decaying memory property is the notorious phenomenon in recurrent neural networks which prohibits the scale up of the models in terms of the sequence length [14, 26].

## 6    Discussion

In Table 1 we compare the classical sequence models including RNN, TCN and attention-based transformer. The state-space model can be considered as an enhancement of Recurrent Neural Networks (RNNs) due to its superior optimization and inference speed. Despite maintaining a similar network topology, inference cost, and memory pattern, it provides a more efficient and streamlined approach. The effort to extend the long-memory learning is also carried out in convolutional networks and attention-based transformers. Romero et al. [22] proposes to parameterize the convolution

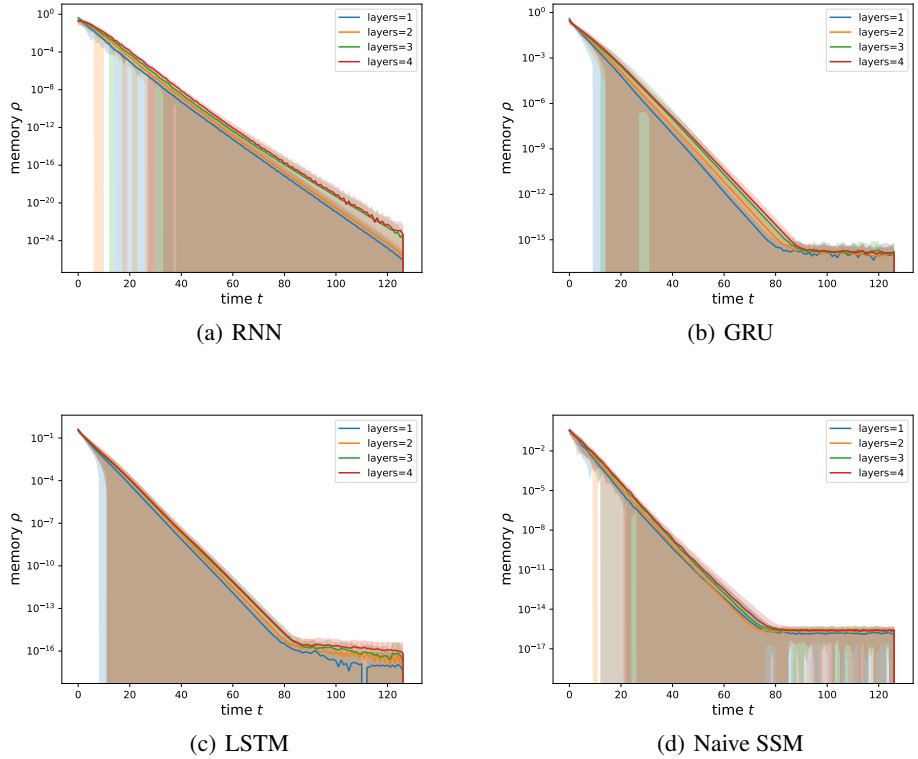

Figure 5: Memory functions of a randomly initialized recurrent networks. The shadow indicates the error bar for 100 repeats.

kernel implicitly, which utilizes the power of spline function approximation $y = \sin(w_0(Wx + b))$. Similar idea has been adopted in Poli et al. [12]. To summarize, we regard SSM, CKConv and linear transformer as the sequence models' improvement in learning long-memory for long sequences.

Table 1: Comparison of sequence models: $T$ is the sequence length, $L$ is the number of layers, $m$ is the hidden dimension, $C$ and $K$ are the total number of channels and convolution kernel sizes in TCN. Input and output dimensions are $d_{\text{input}}$ and $d_{\text{output}}$. Despite TCN's inference cost is independent of sequence length, it depends on the kernel size $K$, which typically bears a similar scale to $T$.

|  | RNN | TCN | Transformer |
|---|---|---|---|
| Number of weights | $Lm(m + d_{\text{input}} + d_{\text{output}})$ | $LCKd_{\text{input}}$ | $3Lmd_{\text{input}}$ |
| Single-step inference cost | $O(1)$ | $O(1)$ | $O(T^2)$ |
| Memory pattern | exponential decay | low rank | no restriction |
| Memory improved version | SSM | CKConv | Linear Transformer |

## 7  Conclusion

In this paper, we give a constructive proof for the universal approximation property of multi-layer state-space models. It is shown that the nonlinear recurrent activations in classical recurrent neural networks are not necessary when there are nonlinear activations across different hidden layers. This result implies state-space model is as powerful as the classical recurrent neural networks in the approximation sense. Furthermore, we study the memory decay in multi-layer state-space models, which is a notorious issue in classical recurrent neural networks. While empirical evidence suggests

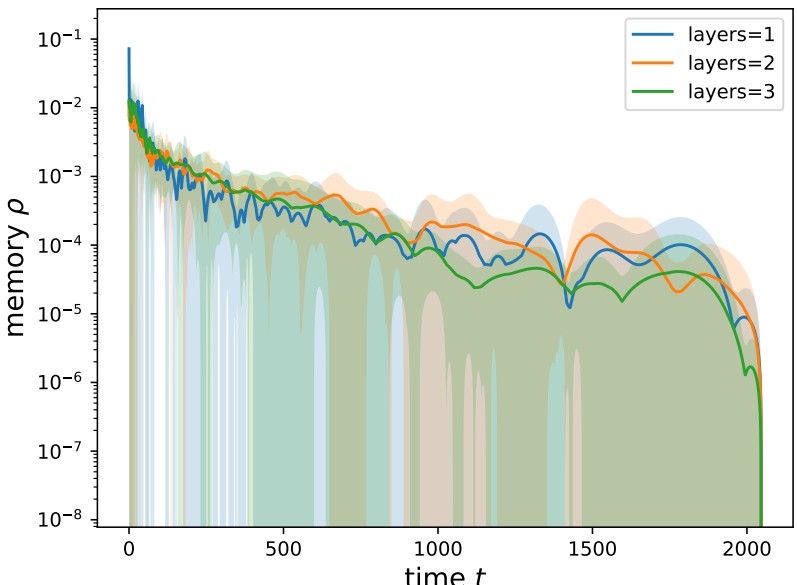

Figure 6: Memory function of a randomly initialized S4. For each model, as the time increases, the memory function can be "capped" by a straight line, which indicate that the memories are decaying exponentially. Compared with Figure 5(d), the results indicate that the smart initialization from S4 provides the memory function with a slower decay.

that state-space models do not experience significant memory decay, they nonetheless exhibit a memory pattern that decays exponentially in the asymptotic sense.

Our research has exciting implications for future work in state-space models. By extending our work to the approximation rate of state-space models, we can obtain better understanding of state-space models' hypothesis space. Such result is important to further optimize the architecture in various real-world applications. We aim to unlock the full potential of state-space models by identifying the ideal network structure (including depth and hidden dimension) for specific tasks and applications.

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

## A  Memory of multi-layer linear RNNs

In this section, we study the memory function of linear RNN. The depth of linear RNN does not directly expand the memroy pattern as the memory are still decaying exponentially. Nonetheless, although the depth does not increase the approximation capacity, a deeper model is endowed with more structural property.

Take the two-layer Linear RNN as an example:

$$\hat{y}_t = \int_0^t Ce^{W_1(t-s)}U_1 \int_0^s e^{W_2(s-r)}U_2 x_r dr ds \tag{27}$$

$$= \int_0^t CP_1 e^{\Lambda_1(t-s)}P_1^{-1}U_1 \int_0^s P_2 e^{\Lambda_2(s-r)}P_2^{-1}U_2 x_r dr ds \tag{28}$$

$$= \int_0^t Ce^{\Lambda_1(t-s)}U_1 \int_0^s e^{\Lambda_2(s-r)}U_2 x_r dr ds \tag{29}$$

$$= \int_{0<r<s<t} dr ds \left( Ce^{\Lambda_1(t-s)}U_1 e^{\Lambda_2(s-r)}U_2 x_r \right) \tag{30}$$

$$= \int_{0<r<t} \int_{r<s<t} dr ds \left( Ce^{\Lambda_1(t-s)}U_1 e^{\Lambda_2(s-r)}U_2 x_r \right) \tag{31}$$

$$= \int_0^t dr \int_r^t ds \left( Ce^{\Lambda_1(t-s)}U_1 e^{\Lambda_2(s-r)}U_2 x_r \right) \tag{32}$$

$$= \sum_{i=1}^{d_{h1}} \sum_{j=1}^{d_{h2}} \int_0^t dr \int_r^t ds \left( Ce^{\lambda_{1i}(t-s)}U_{1,ij} e^{\lambda_{2j}(s-r)}U_2 x_r \right) \tag{33}$$

$$= \sum_{i=1}^{d_{h1}} \sum_{j=1}^{d_{h2}} \int_0^t dr \int_r^t ds \left( e^{\lambda_{1i}(t-s)}U_{1,ij} e^{\lambda_{2j}(s-r)}CU_2 x_r \right) \tag{34}$$

$$= \sum_{ij} \int_0^t dr \int_r^t ds \left( e^{\lambda_{1i}(t-s)+\lambda_{2j}(s-r)}U_{1,ij}CU_2 x_r \right) \tag{35}$$

$$= \sum_{ij} \int_0^t dr \left( \int_r^t ds\, e^{(\lambda_{2j}-\lambda_{1i})s} \right) \left( e^{\lambda_{1i}t-\lambda_{2j}r}U_{1,ij}CU_2 x_r \right) \tag{36}$$

$$= \sum_{ij} \int_0^t dr \left( \frac{1}{\lambda_{2j}-\lambda_{1i}}(e^{(\lambda_{2j}-\lambda_{1i})t} - e^{(\lambda_{2j}-\lambda_{1i})r}) \right) \left( e^{\lambda_{1i}t-\lambda_{2j}r}U_{1,ij}CU_2 x_r \right) \tag{37}$$

$$= \sum_{ij} \int_0^t dr \left( \frac{1}{\lambda_{2j}-\lambda_{1i}}(e^{\lambda_{2j}(t-r)} - e^{\lambda_{1i}(t-r)})U_{1,ij}CU_2 x_r \right) \tag{38}$$

$$= \int_0^t dr \sum_{ij} \left( \frac{1}{\lambda_{2j}-\lambda_{1i}}(e^{\lambda_{2j}(t-r)} - e^{\lambda_{1i}(t-r)})U_{1,ij}CU_2 \right) x_r \tag{39}$$

The memory function of two-layer linear RNN is

$$\hat{\rho}_t = \sum_{ij} \left( \frac{1}{\lambda_{2j}-\lambda_{1i}}(e^{\lambda_{2j}(t)} - e^{\lambda_{1i}(t)})U_{1,ij}CU_2 \right). \tag{40}$$

The main advantage of two-layer linear RNN, compared with single-layer linear RNN, is that it can learn unbounded combination of exponential decay function as long as $\lambda_{2j}$ and $\lambda_{1i}$ are close enough. Notice that for simplicity we do not include the bias term in the evaluation of memory function. The multi-layer linear Recurrent Neural Network (RNN), including a bias term, can be perceived as a linear combination of various depth-specific linear RNNs, devoid of a bias term.

# B  Proof

## B.1  Universal approximation

Here we include the universal approximation from Barron [27], it is frequently used in the later constructive proof for state-space models.

**Theorem B.1** (Cybenko [28], Barron [27]). *For any continous function $f : \mathbb{R}^d \to \mathbb{R}$ and compact set $K \subseteq \mathbb{R}^d$. Assume the activation function $\sigma : \mathbb{R} \to \mathbb{R}$ is bounded and sigmoidal:*

$$\lim_{z \to \infty} \sigma(z) = 1, \ \lim_{z \to -\infty} \sigma(z) = -1. \tag{41}$$

*For any $\epsilon > 0$, there exists $m \in \mathbb{N}$, $A \in \mathbb{R}^{m \times d}$, $b \in \mathbb{R}^k$, $C \in \mathbb{R}^{m \times k}$ such that*

$$\sup_{x \in K} \|f(x) - g(x)\| \le \epsilon, \tag{42}$$

*where*

$$g(x) = C\sigma(Ax + b). \tag{43}$$

## B.2  Proof for Proposition 3.1

*Proof.* Fix the bounded continuous function $f$.

For any $\epsilon > 0$, according to the universal approximation theorem in Theorem B.1, there exists $U_1, U_2, b_1$ such that

$$|f(x) - U_2\sigma(U_1 x + b_1)| \le \epsilon. \tag{44}$$

Take $W_1 = W_2 = b_2 = 0$, the two-layer state-space model can approximate element-wise continuous functions. (Here the element-wise function mean the sequences that for each $k$, $y_k$ only depends on $x_k$.)  □

## B.3  Proof for Proposition 3.2

*Proof.* The original statement is given in the discrete index. We prove the result for the continuous index and the discrete case can be derived by discretization.

Without loss of generality, assume the input output sequence are all one-dimensional. Otherwise we shall stacking the $C, W, U$ for different input-output channels.

Since the input and output are one-dimensional, we know $c, u$ are vectors and $W$ is a square matrix. To approximate the target convolution:

$$y_t = \int_0^t \rho_{t-s} x_s ds \tag{45}$$

by state-space model

$$\hat{y}_t = \int_0^t c^\top e^{W(t-s)} u x_s ds. \tag{46}$$

Since the prediction error can be decomposed into following form

$$y_t - \hat{y}_t = \int_0^t \left( \rho_{t-s} - c^\top e^{W(t-s)} u \right) x_s ds. \tag{47}$$

Approximating the temporal convolution with state-space model over all bounded inputs **x** can be reduced to function approximating problem:

$$\max_{s \in [0,T]} \left| \rho_s - \sum_{i=1}^m c_i e^{-\lambda_i s} \right| < \epsilon, \quad \lambda_i \ge 0. \tag{48}$$

In other words, approximating a convolution layer with state-space model is equivalent to approximating a general integrable function by function with exponential form $\hat{\rho}_s = \sum_{i=1}^m c_i e^{-\lambda_i s}$.

By change of variable, we have

$$\max_{\tau \in [e^{-T}, 1]} |f(\tau) - \sum_{i=1}^{m} c_i \tau^{\lambda_i}| < \epsilon. \tag{49}$$

Here $f(\tau) = \rho_{-\log(\tau)}$. Since polynomials are universal approximators on compact intervals, we know there exists $c_i, \lambda_i$ such that the aforementioned inequality is satisfied. (For example, if $f$ is smooth, we can take the Taylor expansion of function $f$.) □

### B.4 Proof for Proposition 3.6

*Proof.* Our proof is based on Equation (18), for any sequence relationship $H : \{x_k\}_{k=1}^T \to \{y_k\}_{k=1}^T$, we need to approximate $\phi$ and $\Phi$, and subsequently approximate the representation prescribed by the Kolmogorov-Arnold theorem.

Fix tolerance $\epsilon$.

As is shown in Figure 3, the first element-wise function $\phi(\cdot)$ can be approximated by two-layer state-space model. This is a result from the direct application of Proposition 3.1. In math terms, there exists $U_1, U_2, b_1$ such that two-layer state-space model approximate function $\phi$.

$$\|\phi(x) - U_2 \sigma(U_1 x + b_1)\| \leq \epsilon. \tag{50}$$

We shall denote $U_2 \sigma(U_1 x + b_1)$ by $\hat{\phi}(x)$.

Next, according to Proposition 3.2, we know the (temporal) convolution can be approximated via single-layer state-space model. There exists weights $C_3, W_3, U_3$ such that

$$\left\| \sum_{p=1}^{T} \rho_{T-p} \hat{\phi}(x_p + qa) - \sum_{p=1}^{T} C_3 W_3^{T-p} U_3 \hat{\phi}(x_p + qa) \right\| \leq \epsilon. \tag{51}$$

The last layer is again an element-wise function $\Phi(\cdot)$, which can be approximate by two-layer state-space model with weights $U_4, U_5, b_4$. (Figure 3)

$$\left\| \Phi(\sum_{p=1}^{T} C_3 W_3^{T-p} U_3 \hat{\phi}(x_p + qa)) - U_5(U_4(\sum_{p=1}^{T} C_3 W_3^{T-p} U_3 \hat{\phi}(x_p + qa)) + b_4) \right\| \leq \epsilon. \tag{52}$$

Based on the above result, without loss of generality we assume the outer function $\Phi$ is Lipschitz continuous with coefficient $L$, then the final approximation error is bounded by $L\|\rho\|_{L^1}\epsilon + L\epsilon + \epsilon$.

$$\left\| \Phi(\sum_{p=1}^{T} \rho_{T-p} \phi(x_i + qa)) - U_5(U_4(\sum_{p=1}^{T} C_3 W_3^{T-p} U_3 \hat{\phi}(x_p + qa)) + b_4) \right\| \tag{53}$$

$$\leq \left\| \Phi(\sum_{p=1}^{T} \rho_{T-p} \phi(x_i + qa)) - \Phi(\sum_{p=1}^{T} \rho_{T-p} \hat{\phi}(x_p + qa)) \right\| \tag{54}$$

$$+ \left\| \Phi(\sum_{p=1}^{T} \rho_{T-p} \hat{\phi}(x_p + qa)) - \Phi(\sum_{p=1}^{T} C_3 W_3^{T-p} U_3 \hat{\phi}(x_p + qa)) \right\| \tag{55}$$

$$+ \left\| \Phi(\sum_{p=1}^{T} C_3 W_3^{T-p} U_3 \hat{\phi}(x_p + qa)) - U_5(U_4(\sum_{p=1}^{T} C_3 W_3^{T-p} U_3 \hat{\phi}(x_p + qa)) + b_4) \right\| \tag{56}$$

$$\leq L\|\rho\|_{L^1}\epsilon + L\epsilon + \epsilon. \tag{57}$$

To summarize, we achieve the approximation of general nonlinear sequence-to-sequence relationship with representation (with Lipschitz continuous $\Phi$) via five-layer state-space model.

Since the Lipscthiz continuous function is dense in the set of continuous function, therefore the universal approximation can be generalized to the representation with $\Phi$ not necessarily Lipschitz continuous. □

## B.5 Proof for Proposition 3.9

*Proof.* For simplicity, we will only present the approximation of an $n$-th order component of the Volterra Series. The general approximation of nonlinear sequence-to-sequence relationship can be achieved by approximating different order component separately and taking the linear combination.

Consider the target functional

$$y_n(t) = \int_0^t \cdots \int_0^t h_n(\tau_1, \ldots, \tau_n) \prod_{j=1}^n x(t - \tau_j) d\tau_j. \tag{58}$$

The state-space model's $n$-th order term can be represented by

$$\hat{y}_n(t) = \sum_{i=1}^m \int_0^t \cdots \int_0^t \left( \prod_{j=1}^n \hat{h}_j^{(m)}(\tau_j) \right) \prod_{j=1}^n x(t - \tau_j) d\tau_j. \tag{59}$$

It can be seen the kernel function of state-space model is $\hat{h}_n(\tau_1, \ldots, \tau_n) = \sum_{i=1}^m \prod_{j=1}^n \hat{h}_j^{(m)}(\tau_j)$ while the original $n$-th order kernel is multi-variable function $h_n(\tau_1, \ldots, \tau_n)$. For any tolerance $\epsilon$, there exists a sufficiently large hidden dimension $m$ such that the multi-variable function is approximated by the single-variable function's product:

$$\left| \sum_{i=1}^m \prod_{j=1}^n \hat{h}_j^{(m)}(\tau_j) - h_n(\tau_1, \ldots, \tau_n) \right| \le \epsilon. \tag{60}$$

For example, we may consider $\hat{h}_n$ to be polynomial of $\tau_j$ and take the Taylor expansion of kernel $h_n$. □

*Remark* B.2. While the aforementioned proof for approximation is provided through a polynomial method, it's commonly accepted that polynomials may not be the most efficient means to parameterize kernel functions. Consequently, a variety of techniques have been empirically investigated to represent convolutional layers [22].

## B.6 Proof for Proposition 3.11

*Proof.* As $x_t$ converges to $x^*$ exponentially and $\rho$ is integrable, we know the limit of $y_t$ exists

$$\lim_{t \to \infty} y_t = \lim_{t \to \infty} \int_0^\infty \rho_s x_{t-s} ds = \int_0^\infty \rho_s \lim_{t \to \infty} x_{t-s} ds = \int_0^\infty \rho_s x^* ds. \tag{61}$$

Here $\rho_s = C^\top e^{Ws} U, s \ge 0$ is the memory function for (continuous) state-space model.

By the dominated convergence theorem, as $\rho$ is integrable and $e^{c_0 t} |x_{t-s} - x^*|$ is a bounded sequence,

$$\lim_{t \to \infty} e^{c_0 t} \|y_t - y^*\| = \lim_{t \to \infty} e^{c_0 t} \left\| \int_0^\infty \rho_s (x_{t-s} - x^*) ds \right\| \tag{62}$$

$$\le \left\| \int_0^\infty |\rho_s| \lim_{t \to \infty} e^{c_0 t} |x_{t-s} - x^*| ds \right\| \tag{63}$$

$$= \left\| \int_0^\infty |\rho_s| \cdot 0 ds \right\| = 0. \tag{64}$$

□

## B.7 Proof for Proposition 3.12

*Proof.* Since $\lim_{t \to \infty} x_t = x^*$, by continuity of the activation function $\sigma(\cdot)$

$$\lim_{t \to \infty} y_t = \lim_{t \to \infty} \sigma(x_t) = \sigma(\lim_{t \to \infty} x_t) = \sigma(x^*). \tag{65}$$

Hereafter we define $y^* = \sigma(x^*)$.

Since $\sigma$ is Lipschitz continuous, let $L$ be its Lipschitz constant

$$\lim_{t\to\infty} e^{c_0 t}\|y_t - y^*\| = \lim_{t\to\infty} e^{c_0 t}\|\sigma(x_t) - \sigma(x^*)\| \tag{66}$$

$$\leq \lim_{t\to\infty} e^{c_0 t} L\|x_t - x^*\| \tag{67}$$

$$= L \lim_{t\to\infty} e^{c_0 t}\|x_t - x^*\| \tag{68}$$

$$= 0. \tag{69}$$

$\square$

The Lipschitz continuity is a weak assumption that most of the activations such as ReLU, GeLU, tanh, hardtanh satisfy.

## C Limitations

Theoretical results on approximation do not have quantitative results. Therefore it is not easy to directly compare the efficiency of different types of models over a specific task. The good news is that the general memory function servers as a guideline in constructing models. Take the Hyena [12] architecture as an example, as the implicit representation of convolution kernel is utilized in the network, Hyena is not a recurrent model. If we replace the convolution layer by state-space model, we can "translate" a non-recurrent neural network into a recurrent neural network without sacrificing the **approximation capacity**. The main advantage of recurrent networks, compared with implicit convolution form, lies in the lower inference memory cost.

This study primarily explores the qualitative attributes of the state space model within the context of approximation theory. To better delineate it from other architectures like linear transformers, a more in-depth investigation into the rate of approximation is necessary.

## D Comparisons between RNNs, SSMs and S4

In Table 2, we compare different recurrent models in terms of the recurrence, universality, temporal parallel and the memory decay speed.

Our core findings are established for SSM; however, we argue that they are applicable to S4 as well. The primary differences between the vanilla state-space models (SSMs) and S4 involve model parameterisation, weight initialisation, discretisation, normalisation, dropout, and residual connections. The model architectures of SSMs and S4 are almost identical, as both alternately stack linear RNNs and nonlinear activation layers. Therefore, in terms of universal approximation, the approximation capacities of both SSMs and S4 are equivalent.

| | Linear RNN | Nonlinear RNN | SSMs | S4 |
|---|---|---|---|---|
| Recurrence | Yes | Yes | Yes | Yes |
| Universality | No | Yes (hardtanh, tanh) | Yes (ours) | Yes (ours) |
| Temporal Parallel | Yes | No | Yes | Yes |
| Exponential decay | Yes [14] | Yes [16] | Yes (ours) | Yes, moderated (ours) |

Table 2: Comparisons between RNNs, SSMs and S4

## E Further discussion on the two proofs for universality of SSMs

We provided comparison between the two proof methods: An analogy between Kolmogorov-theorem and classical fully-connected neural networks can be drawn because of the finite number of layers in both. It is shown in Eldan and Shamir [29] that a simple function expressible by a small 3-layer feed-forward neural networks cannot be approximated to a certain accuracy unless the network's width is exponential in the dimension. In contrast, Volterra-Series shares similarities with deep learning, and as a result, the advantages of deep learning over classical fully-connected neural networks carry over

to this approach. The authors are inclined to consider Volterra-Series based construction as relatively superior, and increasing the depth to be a more efficient way to scale up the model.

