# OpenReview forum: "State-space models with layer-wise nonlinearity are universal approximators with exponential decaying memory"
_NeurIPS.cc/2023/Conference — NeurIPS 2023 poster_

### Official Review · Reviewer_nePh · 2023-07-06

**Soundness:** 3 good
**Presentation:** 1 poor
**Contribution:** 3 good
**Rating:** 6
**Confidence:** 2

**Summary:**

The paper provides a constructive proof that state-space models (SSMs) are universal approximators of sequence-to-sequence mappings.
Moreover, it shows that SSMs (and even empirically S4) suffer from exponentially decaying memory just like standard RNNs.

**Strengths:**

Up to my knowledge this is the first (constructive) proof that SSMs are universal. I find it in particular interesting that SSMs also suffer from exponentially decaying memory (given the provided definition of such).

**Weaknesses:**

Regrettably, the paper exhibits a significant lack of quality in its writing, containing numerous grammatical errors and typos. It is imperative that it undergoes a comprehensive proofreading process to address these issues.

From my understanding, the paper proves universality for SSMs, which does not include universality for S4 (i.e., SSMs with specific structure of the matrices). This is a major limitation, as in practice no one uses simple SSMs, but only S4. Can S4 be included in the proof? I presume one has to show that also S4 can approximate element-wise functions up to any precision.

The section about the curse of memory is very hard to follow. Can you please rewrite it and properly introduce the concept of memory functions?

The paper is not rigorously written. For instance, it would be much more readable, if the authors would define every variable, e.g., $W \in \mathbb{R}^{m\times m}$ and so on. This would also help the authors avoid using variables that haven't been introduced before.

The provided full paper in the appendix and the main paper differ: for instance in equation 9: $\mathbf{H}$ depends on $t$ in the full paper, but not in the main paper.

Form what I understood, the universality proof of proposition 3.6 approximates functions, where the target value at every index depends on all elements in the full input sequence. However, this is not realistic, as mainly causal operators between sequences have to be learned, i.e., operators that are independent of the future values of the input sequence at every index. Can you please change the proof accordingly?

**Questions:**

See weaknesses

**Limitations:**

The paper provides an interesting and important proof that SSMs are universal, as well as that the memory functions decay exponentially. However, the proof does not include S4 models. Moreover, the quality of writing is very poor, which makes it hard to follow. Overall, the provided proof is very simple, which makes it appealing.

---

> ### Author Rebuttal · Authors · 2023-08-08
>
> We sincerely appreciate your important questions regarding our proof, as well as your constructive suggestions for enhancing the writing quality. Below is our response to the points raised:
>
> 1. "Regrettably, the paper exhibits a significant lack of quality in its writing, containing numerous grammatical errors and typos. It is imperative that it undergoes a comprehensive proofreading process to address these issues."
>
> - Thank you for bringing the writing concern to our attention. We have undertaken thorough proofreading and incorporated the necessary updates in the final revision.
>
> 2. "From my understanding, the paper proves universality for SSMs, which does not include universality for S4 (i.e., SSMs with specific structure of the matrices). This is a major limitation, as in practice no one uses simple SSMs, but only S4. Can S4 be included in the proof? I presume one has to show that also S4 can approximate element-wise functions up to any precision."
>
> - We have prepared a table in the 'Response to all Reviewer' section to compare SSMs with linear RNNs, nonlinear RNNs, and S4. The primary differences between the vanilla state-space models (SSMs) and S4 involve model parameterisation, weight initialisation, discretisation, normalisation, dropout, and residual connections. However, with the exception of the residual connection, the model architectures of SSMs and S4 are almost identical, as both alternately stack linear RNNs and nonlinear activation layers. Therefore, in terms of universal approximation, the approximation capacities of both SSMs and S4 are equivalent.
>
> 3. "The section about the curse of memory is very hard to follow. Can you please rewrite it and properly introduce the concept of memory functions?"
>
> - We have rewritten the section on the curse of memory to improve the introduction of the memory function, as well as the phenomenon of the curse of memory. Due to word constraints in the reply section, the revised fragments are presented in the 'Response to all Reviewer' under the section 'Definition of Memory Function and How to Evaluate It'.
>
> 4. "The paper is not rigorously written. For instance, it would be much more readable, if the authors would define every variable, e.g., $W \in \mathbb{R}^{m\times m}$ and so on. This would also help the authors avoid using variables that haven't been introduced before."
>
> - Thank you for pointing this out. We have updated variable dimension in SSM single layer general form as follows: $x \in \mathbb{R}^{d_{in}}, y \in \mathbb{R}^{d_{out}}, h \in \mathbb{R}^m, W \in \mathbb{R}^{m \times m}, U \in \mathbb{R}^{m \times d_{in}}, b \in \mathbb{R}^m, C \in \mathbb{R}^{d_{out} \times m}, D \in \mathbb{R}^{d_{out} \times d_{in}}$.
>
> 5. "The provided full paper in the appendix and the main paper differ: for instance in equation 9: $\mathbf{H}$ depends on $t$ in the full paper, but not in the main paper."
>
> - Thank you for the careful reading and pointing out the difference. We discovered the missing subscript $t$ and added it in the full paper.
>
> 6. "Form what I understood, the universality proof of proposition 3.6 approximates functions, where the target value at every index depends on all elements in the full input sequence. However, this is not realistic, as mainly causal operators between sequences have to be learned, i.e., operators that are independent of the future values of the input sequence at every index. Can you please change the proof accordingly?"
>
> - Thank you so much for raising this important question. In proposition 3.6, we did not explicitly specify assumptions for the targets other than the **continuity** w.r.t. the inputs. Here we discuss several other important properties expected from the targets. In the context of proposition 3.9, we explore these properties as defined in paper [1] Definition 3.1.
>     1. **Causality**. Autoregressive problems such as language modelling focus on the cases that output $y_t$ only depend on the inputs $x_{0, \dots, t}$. The causality of the targets guarantees the temporal convolution kernels $\rho_k$ in Figure 4 having the property $\rho_k (t) = 0, t \leq 0.$ This causal temporal convolution, together with proposition 3.2, ensures that state-space model can be implemented in a recurrent form.
>     2. **Time-homogeneous**. In sequence modelling, the concept of being time-homogeneous is fundamental. This is because time-inhomogeneous targets may not be approximated to an arbitrary level of accuracy, regardless of how much data or computational resources are available.
>     3. **Bounded & Regular**. These two properties are usually assumed from the theoretical consideration. As shown in [1], continuous causal regular and time-homogeneous linear functionals have a convolution form based on the Riesz representation.
>
> ### **Limitations:**
>
> 7. "The paper provides an interesting and important proof that SSMs are universal, as well as that the memory functions decay exponentially. However, the proof does not include S4 models. Moreover, the quality of writing is very poor, which makes it hard to follow. Overall, the provided proof is very simple, which makes it appealing."
>
> - As mentioned in response to the second comment, the primary differences between SSMs and S4 models lie in aspects other than the model architecture. Therefore, the universality proof for SSMs should also naturally extend to S4 models.
> - We apologise for the current state of our manuscript's writing. We are dedicated to improving this aspect significantly in the final revision.
> - We are glad to know that you found our proof's simplicity appealing. Our aim is to maintain rigour while ensuring comprehensibility.
>
> ---
> [1] Zhong Li,  Jiequn Han, E. Weinan, and Qianxiao Li. "On the Curse of Memory in Recurrent Neural Networks: Approximation and Optimization Analysis." In International Conference on Learning Representations. 2020.

---

> > ### Comment · Reviewer_nePh · 2023-08-17
> > **Thanks for the detailed response**
> >
> > Thanks for the detailed response. The promised changes by the authors will increase the quality of the paper. I thus increase my rating.

---

### Official Review · Reviewer_QCvt · 2023-07-06

**Soundness:** 2 fair
**Presentation:** 4 excellent
**Contribution:** 1 poor
**Rating:** 6
**Confidence:** 3

**Summary:**

This paper attempts to show several properties of SSMs: a) SSMs can approximate element-wise functions and temporal convolutions b) Universality of SSMs c) Exponential memory decay. The authors also show experiments of SSM's exponential memory decay and compare them against variants of RNN architectures.

**Strengths:**

* Thorough background section and clear introduction of the RNN and SMM architecture.

* Clear writing.

* The usage of the Kolmogorov-Arnold thoerem and Volterra series is interesting.


**Weaknesses:**

* My main concern is most of the main results seems trivial: When taking away the tricks that makes SSMs work such as smarter initialization and diagonal parameterisation, architecture wise, SSMs are simply an RNN without non-linearly between hidden states. Hence, the theory results (i.e. proposition 3.1, 3.2, 3.12 and theorem 3.13) seems to transfer very trivially from RNNs.

*  The 2 formulations for showing universality of SMMs in section 3.2 seems to deviate greatly from the SSM architecture (i.e. figure 3 and 4 does not follow the definition of SSMs in equation 1 and 2). It is unclear how much the result transfers for standard SSMs used in practice.

* In figure 5 and 6 there are no error bars or mentioning of number of seeds.



**Questions:**

Authors may address my comments above.

---

> ### Author Rebuttal · Authors · 2023-08-09
>
> We sincerely appreciate your helpful comments for us to provide clarifications on the SSM architecture, its relation to RNNs, and therefore the significance of our findings. Below is our response to the points raised:
>
> 1. "My main concern is most of the main results seems trivial: When taking away the tricks that makes SSMs work such as smarter initialisation and diagonal parameterisation, architecture wise, SSMs are simply an RNN without non-linearly between hidden states. Hence, the theory results (i.e. proposition 3.1, 3.2, 3.12 and theorem 3.13) seems to transfer very trivially from RNNs."
>
> -  SSM is similar to RNN in the sense that it stacks linear RNN and layer-wise nonlinearity alternately. **However, SSMs are different from linear RNNs as linear RNNs are not universal approximators. While general nonlinear RNNs are universal, they introduce nonlinearity differently, leading to distinct dynamics between SSMs and nonlinear RNNs.** Therefore SSMs are different from linear RNNs and traditional nonlinear RNNs. In hindsight, the universal approximation and exponential memory decay might seem simple. However, these results shall contribute to the understanding of SSMs' excellent performance over various long-sequence modelling.
> - Since proposition 3.1 is the direct application of universal approximation property of one hidden-layer neural network and proposition 3.2 is the inherent property of linear RNN, they might seem simple from the proofs. However, these two propositions are the indispensable building blocks to achieve the main results (proofs for universality of SSMs) in proposition 3.6 and proposition 3.9. The exponential memory decay results in proposition 3.12 and theorem 3.13 show that layer-wise nonlinearity does not change the qualitative memory pattern of recurrent models.
>
> 2. "The 2 formulations for showing universality of SSMs in section 3.2 seems to deviate greatly from the SSM architecture (i.e. figure 3 and 4 does not follow the definition of SSMs in equation 1 and 2). It is unclear how much the result transfers for standard SSMs used in practice."
>
> - Equations 1 and 2 only illustrate the structure of single-layer state-space models. The constructions in Propositions 3.6 and 3.9 for multi-layer state-space models correspond to the structures demonstrated in Figures 3 and 4, respectively. **They show that by alternating between stacking simple linear RNNs and applying layer-wise nonlinearity, universality can be achieved.** **In practice, multi-layer state-space models are commonly used for various sequence modelling tasks (e.g., 2~32 layers in [1], 6 layers in [2], 4 layers in [3])**. Additionally, a single-layer state-space model without nonlinear activation is not universal and cannot be used to learn general nonlinear sequence-to-sequence relationships. Consequently, our theory characterises the standard SSM models prevalent in practical use.
>
> 3. "In figure 5 and 6 there are no error bars or mentioning of number of seeds."
>
> - Thank you for bringing this oversight to our attention. Error bars and seed counts are indeed critical for interpreting experimental results. We have updated figures 5 and 6 to include error bars. The experiments are repeated 100 times to obtain the error bars.
>
> ---
> [1] Albert Gu, Karan Goel, and Christopher Re. "Efficiently Modeling Long Sequences with Structured State Spaces." In _International Conference on Learning Representations_. 2021.
>
> [2] Eric Nguyen, Karan Goel, Albert Gu, Gordon Downs, Preey Shah, Tri Dao, Stephen Baccus, and Christopher Ré. "S4nd: Modeling images and videos as multidimensional signals with state spaces." _Advances in neural information processing systems_ 35 (2022): 2846-2861.
>
> [3] Albert Gu, Karan Goel, Ankit Gupta, and Christopher Ré. "On the parameterization and initialization of diagonal state space models." _Advances in Neural Information Processing Systems_ 35 (2022): 35971-35983.

---

> > ### Comment · Reviewer_QCvt · 2023-08-17
> > **Official Comment**
> >
> > Thank you authors, for your detailed response, I have adjusted the score accordingly

---

### Official Review · Reviewer_gZh5 · 2023-07-06

**Soundness:** 3 good
**Presentation:** 3 good
**Contribution:** 4 excellent
**Rating:** 8
**Confidence:** 4

**Summary:**

This paper analyzes properties of state space models (SSMs) and verifies the analytic results with numerical simulations.

The primary result is that SSMs have the same basic properties as classic RNNs.  They perform temporal convolution on their inputs, with exponentially decaying memory and are universal function approximators.

**Strengths:**

My overall assessment is that this paper is an important contribution to our understanding of SSMs.  It raises really important questions about how SSMs are apparently able to learn long-range dependencies and points to the generality of exponentially-decaying memory.

The exposition was pretty clear; although the math is challenging for a broad audience, the simulations give confidence that the results are sound.

**Weaknesses:**

It's worth thinking more about presentation of the figures.  Figure 4 was pretty helpful but Figure 3 was pretty baffling.   Generally speaking my suggestion is that the captions could be more explicit and all of the terms in the expressions should appear in the figures.

The paper raises a really important question that should be addressed more explicitly.  If SSMs have basically the same properties as generic RNNs, why do they apparently work so much better?  I appreciate that it may not be possible to answer this question with certainty, but some thoughtful discussion would really enhance the impact of this paper.

**Questions:**

Why do SSMs work well empirically?  Is it just that they more easily find long time constants in their exponential decay?   I think it's really important to say something substantive about this.

**Limitations:**

Why do SSMs work well?

---

> ### Author Rebuttal · Authors · 2023-08-09
>
> We sincerely appreciate your positive review and insightful questions regarding the reasons behind the outstanding performance of SSMs. Below is our response to the points raised:
>
> ### **Weaknesses:**
>
> 1. "It's worth thinking more about presentation of the figures. Figure 4 was pretty helpful but Figure 3 was pretty baffling. Generally speaking my suggestion is that the captions could be more explicit and all of the terms in the expressions should appear in the figures."
>
> - We greatly appreciate your helpful feedback regarding the presentation of our figures. We have enhanced Figure 3 by revising its captions and incorporating the necessary equations into the figure itself. Please let us know if this addresses your question or if you have any other feedback.
>
> 2. "The paper raises a really important question that should be addressed more explicitly. If SSMs have basically the same properties as generic RNNs, why do they apparently work so much better? I appreciate that it may not be possible to answer this question with certainty, but some thoughtful discussion would really enhance the impact of this paper."
>
> - Since both of SSM and RNN have universal approximation capabilities, their difference in performance mainly stems from the training process. Unlike traditional RNNs with recurrent nonlinearity, the linear RNN structure of SSM significantly reduces the time cost of forward and backward operations, scaling it down from $O(T)$ to $O(\log T)$. While RNNs such as [4] used a sequence length of 100, the fast training speed of SSM helps handle longer input sequences. Besides training speed, there could also be some other underlying reasons that require further ablation studies to pinpoint.
>
> ### **Questions:**
>
> 3. "Why do SSMs work well empirically? Is it just that they more easily find long time constants in their exponential decay? I think it's really important to say something substantive about this."
>
> - Thank you for the inspiring question. We will incorporate the following discussion into the revision.
> - In response to the inquiry on why SSMs (or specifically S4) perform well empirically, there are several dimensions to consider. Primarily, both SSMs and S4s possess a universal approximation property that forms the theoretical foundation of their numerical performance. Secondly, the parameterisation of the recurrent matrix and the time discretisation in S4 maintains stability as elucidated in [1]. This stability ensures the model's capability in approximating long-memory targets, even when the model's memory undergoes exponential decay. Thirdly, SSMs benefit from the parallelisation capabilities of linear RNNs through techniques like the fast Fourier transform or associative scan, which scale the time complexity down from $O(T)$ to $O(\log T)$. Moreover, the weight initialisation method illustrated in [2,3] further augments the model's capacity to learn long-term memory. Lastly, the success of S4 can be attributed to the integration of widely used techniques such as layer normalisation, dropout, and residual connections.
>
> ### **Limitations:**
>
> 4. "Why do SSMs work well?"
>
> - In response to the query regarding the efficacy of SSMs, we have delved into an analysis encompassing various aspects such as universality, parameterisation, training methodology, initialisation, and other prevalent techniques, as elucidated in our reply to Question 3. While we acknowledge that our insights might not offer a comprehensive explanation, we are optimistic that our analysis provides a valuable contribution towards the community's evolving understanding of the model.
>
> ---
> [1] Shida Wang,  Zhong Li, and Qianxiao Li. "Inverse Approximation Theory for Nonlinear Recurrent Neural Networks." _arXiv preprint arXiv:2305.19190_ (2023).
>
> [2] Albert Gu, Tri Dao, Stefano Ermon, Atri Rudra, and Christopher Ré. "Hippo: Recurrent memory with optimal polynomial projections." _Advances in neural information processing systems_ 33 (2020): 1474-1487.
>
> [3] Albert Gu,  Karan Goel, and Christopher Re. "Efficiently Modeling Long Sequences with Structured State Spaces." In _International Conference on Learning Representations_. 2021
>
> [4] Jack Rae, Chris Dyer, Peter Dayan, and Timothy Lillicrap. "Fast parametric learning with activation memorization." In _International Conference on Machine Learning_, pp. 4228-4237. PMLR, 2018.

---

> ### Comment · Reviewer_gZh5 · 2023-08-10
>
> Thank you for the thoughtful response.

---

### Official Review · Reviewer_KVQv · 2023-07-06

**Soundness:** 3 good
**Presentation:** 2 fair
**Contribution:** 3 good
**Rating:** 5
**Confidence:** 2

**Summary:**

The authors set out to analyze state space models, which have been gaining popularity as alternatives to transformer based systems that can better model long range dependencies and are more computationally efficient. Since such models do not utilize a non-linear activation function along the temporal access, it is important to analyze if this poses a restriction to their modeling capabilities. The paper provides a construction-based argument to prove that as long as there are layer-wise nonlinearities in the model, it becomes a universal approximator for any sequence model and thus does not require activation functions in the temporal domain to improve model capacity. Further, the authors also show that like the earlier recurrent networks (RNN, GRU, etc.), these models also suffer from an exponentially decaying memory.

While the paper tackles quite a relevant topic on a family of models that are becoming widely popular, the draft itself could use some work. In particular, the presentation and clarity of the work needs to be improved, and it would be very helpful to also get some intuition behind the propositions mentioned and proved. That is, to understand what the propositions imply and why they are important.

**Strengths:**

- The authors show that the state space models (SSMs) which have no non-linearity in their temporal axis can still universally approximate any sequential mapping up to some arbitrary error. This is proved by construction.
- The authors also show that like RNNs, such models also suffer from the exponential memory decay problem.
- Some preliminary numerical experiments are conducted to test the memory decay problem across different recurrent models.

**Weaknesses:**

- The writing of the draft can be substantially improved. It would be good to include some explanations into the implications of each proposition and theorem, as to understand why exactly it is meaningful and important. The plots are currently taking a lot of extra space which can be freed to add this content.
- It would be nice if the authors could provide some kind of rates of memory decay to better understand if it is better or worse in SSMs than other RNN-based methodologies.

**Questions:**

- Line 48: Why is this relationship nonlinear? It looks like a linear function of $x$.
- Line 83: Does the definition of $\hat{\rho}_t$ involve $t$ in the exponential or $(t-s)$ which is written in the draft (mistakingly, I believe?)
- Could the authors provide some clarification and their reasoning on their construction of the memory function? Why is this reasonable? How is this computed?
- For universality results in Section 3.3, could the authors provide some intuition behind Remark 3.4, what it implies and how is it achieved?
- Is there a difference in some assumptions between Kolmogorov-theorem based construction with Volterra-Series based construction? Is the latter always superior? What cost does it come with?

**Limitations:**

The authors have adequately addressed the limitations.

---

> ### Author Rebuttal · Authors · 2023-08-09
>
> We sincerely appreciate your constructive feedback on providing more explanation and intuition behind the mathematical proof.  Below is our response to the points raised:
>
> 1. "The writing of the draft can be substantially improved. It would be good to include some explanations into the implications of each proposition and theorem, as to understand why exactly it is meaningful and important. The plots are currently taking a lot of extra space which can be freed to add this content."
>
> - Thank you for the suggestions on the paper writing. We have incorporated the following implications into the final revision:
> - The implications of Propositions 3.1 and 3.2 are modest yet vital: Proposition 3.1 works on the approximation of element-wise (or layer-wise) nonlinear functions, while Proposition 3.2 focuses on temporal convolution. The findings demonstrate that these two types of operators can indeed be approximated using state-space models. While the proof approach may appear elementary, it plays a pivotal role, laying the groundwork for the more advanced findings presented in Propositions 3.6 and 3.9.
> - Implications of Theorem 3.5 and Proposition 3.6: The Kolmogorov-Arnold theorem shows the feasibility to **decompose any continuous multi-variable function into single-variable functions and addition operators.** In state-space model, the target causal time-homogeneous sequential relationship can be viewed as a multi-variable function where each input is a variable. Since linear RNN provides the addition across different inputs, the element-wise nonlinearity corresponds to the single-variable function.
> - Implications of Theorem 3.8 and Proposition 3.9: The Volterra Series approach is a more deep-learning flavour construction while the Kolmogorov-Arnold theorem provides a classical finite-layer approximation result. See the answer to question 7 on the comparison of two approaches.
> - Implications of Proposition 3.11 and Proposition 3.12: Despite the state-space model's proficiency in managing numerous long-sequence tasks, including those within the long-range arena, it exhibits an inherent exponential memory decay. **Our findings, however, suggest that the smart initialisation from S4 can moderate this issue, resulting in a slower memory decay function.**
>
> 2. "It would be nice if the authors could provide some kind of rates of memory decay to better understand if it is better or worse in SSMs than other RNN-based methodologies."
>
> - Thank you for your valuable suggestion. We acknowledge the importance of understanding how memory decay in SSMs compared to other RNN. However, according to the survey paper [1], there is limited work on the rate of memory rate for different SSMs. Direct comparisons of these rates could be complex, given that they might hinge on multiple factors, including the number of parameters. We consider this as a topic for future exploration.
>
> ### **Questions:**
>
> 3. "Line 48: Why is this relationship nonlinear? It looks like a linear function of $x$."
>
> - Yes, it is a typo. Thank you for pointing this out. We have modified it into `the first component` in the revision.
>
> 4. "Line 83: Does the definition of $\hat{\rho}_t$ involve $t$ in the exponential or $(t-s)$ which is written in the draft (mistakingly, I believe?)"
>
> - Thank you so much for pointing out the mistake in the notation, it should be $\hat{\rho}_{t-s}$. We have modified it in the revision of the paper.
>
> 5. "Could the authors provide some clarification and their reasoning on their construction of the memory function? Why is this reasonable? How is this computed?"
>
> - Due to word limitations, we have moved the reply to this question to the "Response to all Reviewer" section, specifically under the topic "Definition of Memory Function and How to Evaluate it".
>
> 6. "For universality results in Section 3.3, could the authors provide some intuition behind Remark 3.4, what it implies and how is it achieved?"
>
> - We first clarify a typo in the Remark 3.4: `we use the same` $\Phi$. The first Kolmogorov theorem is proved for different $\Phi_q$ and $\phi_{q, p}$. Subsequent studies refine this process, minimising the variety of functions required to achieve the final form of Equation (16). This refinement can be interpreted as an application of **weight sharing** in the approximation of functions $\Phi$ and $\phi$. Essentially, the practice of weight sharing in SSMs is substantiated by the diverse results cited in Remark 3.4.
>
> 7. "Is there a difference in some assumptions between Kolmogorov-theorem based construction with Volterra-Series based construction? Is the latter always superior? What cost does it come with?"
>
> - Thank you so much for your very thoughtful and inspiring question. There is no direct difference in terms of assumptions between Kolmogorov-theorem based construction and Volterra-Series based construction. Both constructions work for causal regular time-homogeneous nonlinear functionals, in other words, sequence relationships.
> - Please refer to "Response to all Reviewers" , under "Highlights of Other Modifications" section, point number 2 for a detailed comparison between the two proof methods.
> - Certainly, the Volterra-Series based construction comes with a cost similar to that of deep learning. Its complex structure makes the analysis of the model dynamics difficult to analyse.
>
> ---
> [1] Haotian Jiang, Qianxiao Li, Zhong Li & Shida Wang. (2023). A Brief Survey on the Approximation Theory for Sequence Modelling. _Journal of Machine Learning_. _2_ (1). 1-30.

---

> > ### Comment · Reviewer_KVQv · 2023-08-16
> > **Official Comment**
> >
> > Thanks for the response as well as the clarifications provided. In light of this, I am inclined to raise my score.

---

### Official Review · Reviewer_rFCc · 2023-07-12

**Soundness:** 3 good
**Presentation:** 3 good
**Contribution:** 2 fair
**Rating:** 6
**Confidence:** 4

**Summary:**

The paper presents universal approximation results for state-space models (SSM) with layer-wise nonlinearity. In particular, two-layer SSMs with layer-wise nonlinearity can approximate any continuous function over a compact set. Moreover, SSMs can approximate elementwise function and convolution. The paper also characterizes the memory behavior of SSMs, showing that their memory still decays exponentially when the eigenvalues of the A matrix is bounded from 1. These theoretical results are validated with numerical experiments.

**Strengths:**

1. Universal approximation results are helpful to theoretically characterize the expressive power of SSMs. These models are becoming more popular and these results are not yet known to the best of my knowledge. Consequently the paper could help better understand these models.

2. The presentation is quite clear. Even though this paper is theory-heavy, the main proof ideas are summarized in the main paper. This helps readers understand the technical challenges and the idea of the approach.

**Weaknesses:**

1. Lacking connection to linear RNNs. SSMs are a special case of linear RNNs, and universal approximation results of linear RNNs are available. Moreover, one can show that almost any linear RNNs can be written as SSMs (e.g. Sec 2.2 of Gupta et al. 2022). This is simply because almost any matrix A can be diagonalized, and put into RNNs. Could the university approximation results of linear RNNs be used to prove university approximation results of SSM through this connection?

2. Lack of applications, or a direction to help with applications. While the theoretical results could help with the understanding of these models, the paper does not point out potentially applications of these theoretical results. This might limit the significance of these results.


[1] Simplifying and Understanding State Space Models with Diagonal Linear RNNs. Ankit Gupta, Harsh Mehta, Jonathan Berant. 2022.

**Questions:**

1. What are the dimensions of input/output of the SSMs? The notation is unclear on this.
2. Eq (9): What's H(x_test). Why is rho_hat defined this way?
More generally I don't understand this section.
3. Why does the paper focus on approximation results for elementwise function and convolution?
Elementwise function has no recurrent so it's all about the layer-wise non-linearity?
4. I don't understand Prop 3.12. Is it the same constant c0?


**Limitations:**

Not necessary.

---

> ### Author Rebuttal · Authors · 2023-08-09
>
> We sincerely appreciate your constructive comments for us to clarify details in our work. Below is our response to the points raised:
> ### **Weaknesses:**
> 1. "Lacking connection to linear RNNs. SSMs are a special case of linear RNNs, and universal approximation results of linear RNNs are available. Moreover, one can show that almost any linear RNNs can be written as SSMs (e.g. Sec 2.2 of Gupta et al. 2022). This is simply because almost any matrix A can be diagonalised, and put into RNNs. Could the university approximation results of linear RNNs be used to prove university approximation results of SSM through this connection?"
>
> - Thank you for your suggestion. We have attached a table comparing linear RNNs, nonlinear RNNs, state-space models, S4 in detail in 'Response to all Reviewer' section.
> - Sec 2.2 of Gupta et al. 2022 claims that `Diagonal Linear RNNs (DLRs) are as expressive as general linear RNNs`. Indeed we may consider diagonal case of SSMs, but it is important to note that the result in Gupta et al. 2022 does not contribute to the universality of linear RNN, and therefore is not directly relevant with the proof of SSMs universality. Moreover, there is more to SSMs than linear RNN that we have taken into consideration, such as the unique architecture of alternating linear and non-linear layers. Therefore, the connection between the two might not be as strong unfortunately.
>
> 2. "Lack of applications, or a direction to help with applications. While the theoretical results could help with the understanding of these models, the paper does not point out potentially applications of these theoretical results. This might limit the significance of these results."
>
> - While universal approximation theorem doesn't guarantee the ease of learning or the speed of convergence in training a network, it does provide a theoretical underpinning for understanding why, given enough data and computational power, neural networks are capable of excellent performance.
> - While it is straightforward that linear RNN can only approximate linear functionals, we proved that state-space models (linear RNN with layer-wise nonlinearity) has the universal approximation property. This property is a theoretical guarantee for various applications. In practice, State Space Models (SSMs) demonstrate performance comparable to transformer models (see [1] [2]). Practitioners often question whether a model's success on one task extends to new tasks. **Our findings support the viability of using SSMs for these new tasks.**
>
> ### **Questions:**
>
> 1. "What are the dimensions of input/output of the SSMs? The notation is unclear on this."
>
> - Thank you for pointing this out. We have updated variable dimension in SSM single layer general form as follows: $x \in \mathbb{R}^{d_{in}}, y \in \mathbb{R}^{d_{out}}, h \in \mathbb{R}^m, W \in \mathbb{R}^{m \times m}, U \in \mathbb{R}^{m \times d_{in}}, b \in \mathbb{R}^m, C \in \mathbb{R}^{d_{out} \times m}, D \in \mathbb{R}^{d_{out} \times d_{in}}$. For multi-layer state-space model, the nonlinear activation is added layer-wise as shown in Figure 1.
>
> 2. "Eq (9): What's H(x_test). Why is rho_hat defined this way? More generally I don't understand this section."
>
> - Thank you for pointing out the typo in the paper. We have modified this in the full paper version in supplementary material as $\hat{y}_t = \mathbf{H}_t (\mathbf{x}^{\textrm{test}})$.
> - For consistency with the memory function $\rho$ in linear functional analysis, we substitute the test input with the Heaviside inputs denoted by $\mathbf{x}^{\textrm{test}}$ where $\mathbf{x}^{\textrm{test}}(t) = x,$ if $t > 0$, and $\mathbf{x}^{\textrm{test}}(t) = 0,$ if $t \leq 0$.
> - Due to the word limitations in individual responses, we have moved our reply to this question to the "Response to all Reviewer" section, specifically under the topic "Definition of Memory Function and How to Evaluate It."
>
> 3. Why does the paper focus on approximation results for elementwise function and convolution? Elementwise function has no recurrent so it's all about the layer-wise non-linearity?
>
> - In our paper, the main result of universal approximation theorem is established in section 3.3 based on Kolmogorov-Arnold representation theorem and Volterra Series. In particular, our proof relies heavily on element-wise nonlinear functions and temporal convolutions - these serve as the primary building blocks in our theoretical structure. Therefore, in the section 3.1 and 3.2, we first establish the feasibility of approximating element-wise nonlinear functions and temporal convolutions with state-space models. While it is true that element-wise functions lack recurrence, they provide nonlinearity without breaking the recurrence. It is important to highlight that the nonlinearity between layers alone is sufficient to achieve universality of state-space models. For example, it can approximate nonlinear RNNs with nonlinear recurrent activations. We are happy to elaborate further on any point.
>
> 4. I don't understand Prop 3.12. Is it the same constant c0?
>
> - As is shown in appendix B.7, it is the same constant $c_0$.
>
> ---
> [1] Albert Gu, Karan Goel, and Christopher Re. "Efficiently Modeling Long Sequences with Structured State Spaces." In International Conference on Learning Representations. 2021.
>
> [2] Jimmy TH Smith, Andrew Warrington, and Scott Linderman. "Simplified State Space Layers for Sequence Modeling." In The Eleventh International Conference on Learning Representations. 2022.

---

> > ### Comment · Reviewer_rFCc · 2023-08-22
> >
> > Thanks to the authors for the explanation. I remain supportive of the paper.

---

### Author Rebuttal · Authors · 2023-08-09

## Response to all Reviewers
We thank all the reviewers for their insightful reviews. The reviewers thought the work tackles **"quite a relevant topic on a family of models that are becoming widely popular"**, provides result that **"is an important contribution to our understanding of SSMs"**, with **"an interesting and important proof"**. The experiments demonstrate a memory decay phenomenon that is **”particular interesting”** and **”give confidence that the results are sound”**.

Below we present a summary of our response and the corresponding major changes.

### Comparisons Between RNNs, SSMs and S4
**Memory decay**: All the recurrent models in the table suffer from exponential decay in memory. However, S4 has a slower decay (as shown in experiments) with a suitable initialisation.

| | Linear RNN| Nonlinear RNN| SSMs| S4|
|---|---|---|---|---|
| Recurrence| Yes| Yes| Yes| Yes|
| Universality| No| Yes (hardtanh,tanh) [1]| Yes (ours) | Yes (ours)|
| Temporal Parallel| Yes| No | Yes| Yes|
| Exponential decay| Yes, [2]| Yes, [3]| Yes (ours) | Yes, moderated (ours)|

### Definition of Memory Function and How to Evaluate It
- As a motivation for the memory function, [2] proves that a bounded causal continuous regular time-homogeneous linear functional has the following Riesz representation: $y_t = \mathbf{H_\mathrm{t}} (\mathbf{x}) = \int_{-\infty}^t \rho_{t-s} x_sds.$ Here $\rho_t$ is an $L_1$ integrable function. If $\rho_t$ rapidly decreases with $t$, then the target sequence map $\mathbf{H}_t$ has a short-term memory. Consequently, we refer to $\rho_t$ as the memory function, since it captures the memory property of a linear functional in its entirety.
- For consistency with the memory function $\rho$ in linear functional analysis, we substitute the original impulse test input $(1, 0, 0, \dots)$ with the Heaviside inputs denoted by $\mathbf{x}^{\textrm{test}}$ where $\mathbf{x}^{\textrm{test}}(t) = x,$ if $t > 0$, and $\mathbf{x}^{\textrm{test}}(t) = 0,$ if $t \leq 0$.
- Notice that the derivative of a linear functional at test input extracts the memory function $|\frac{d}{dt} H_t( \mathbf{x}^{\textrm{test}} ) | = |\rho(t)|_2$. Therefore a natural extension of the memory function $\rho_t$ to the nonlinear functionals is given in Section 2.3.
$\displaystyle \hat{\rho}(t) = |\frac{d \hat y_t}{dt}|_2$ where $\hat{y}_t = \widehat{\mathbf{H}}_t ({\mathbf{x}^{\textrm{test}}}).$
- Memory function can be evaluated by computing the model's derivative at the test input using finite difference method.

### Highlights of Other Modifications
1. We added discussion on **possible reasons for the good performance of SSMs**:
	 Primarily, both SSMs and S4s possess a universal approximation property that forms the theoretical foundation of their numerical performance. Secondly, the parameterisation of the recurrent matrix and the time discretisation in S4 maintains stability as elucidated in [3]. This stability ensures the model's capability in approximating long-memory targets, even when the model's memory undergoes exponential decay. Thirdly, SSMs benefit from the parallelisation capabilities of linear RNNs through techniques like the fast Fourier transform or associative scan, which scale the time complexity down from $O(T)$ to $O(\log T)$. Moreover, the weight initialisation method illustrated in [4,5] further augments the model's capacity to learn long-term memory. Lastly, the success of S4 can be attributed to the integration of widely used techniques such as layer normalisation, dropout, and residual connections.

2. We provided comparison between the two proof methods:
	**An analogy between Kolmogorov-theorem and classical fully-connected neural networks can be drawn because of the finite number of layers in both.** It is shown in [6] that a simple function expressible by a small 3-layer feedforward neural networks cannot be approximated to a certain accuracy unless the network's width is exponential in the dimension. **In contrast, Volterra-Series shares similarities with deep learning**, and as a result, the advantages of deep learning over classical fully-connected neural networks carry over to this approach. **The authors are inclined to consider Volterra-Series based construction as relatively superior, and increasing the depth to be a more efficient way to scale up the model.**

3. Dimension of variables are attached to the paper, typos and grammar errors are resolved.
---
[1] Lukas Gonon and Juan-Pablo Ortega. "Reservoir computing universality with stochastic inputs." IEEE transactions on neural networks and learning systems 31, no. 1 (2019): 100-112.

[2] Zhong Li, Jiequn Han, E. Weinan, and Qianxiao Li. "On the Curse of Memory in Recurrent Neural Networks: Approximation and Optimization Analysis." In International Conference on Learning Representations. 2020.

[3] Shida Wang, Zhong Li, and Qianxiao Li. "Inverse Approximation Theory for Nonlinear Recurrent Neural Networks." _arXiv preprint arXiv:2305.19190_ (2023).

[4] Albert Gu, Tri Dao, Stefano Ermon, Atri Rudra, and Christopher Ré. "Hippo: Recurrent memory with optimal polynomial projections." _Advances in neural information processing systems_ 33 (2020): 1474-1487.

[5] Albert Gu, Karan Goel, and Christopher Re. "Efficiently Modeling Long Sequences with Structured State Spaces." In _International Conference on Learning Representations_. 2021

[6] Ronen Eldan and Ohad Shamir. "The power of depth for feedforward neural networks." In Conference on learning theory, pp. 907-940. PMLR, 2016.

---

### Decision · Program_Chairs · 2023-09-21

**Decision:**

Accept (poster)

**Comment:**

This paper provides theory for sequence-to-sequence models based on state space layers, which have achieved state-of-the-art performance on a challenging benchmarks. The reviewers agree that the universal approximation and exponential memory decay results are valuable contributions to the field, and many concerns were addressed through the discussion period.